# Variational Auto-encoders for Causal Inference

## Abstract

Estimating causal effects from observational data (at either an individual- or a population-level) is critical for making many types of decisions. One approach to address this task is to learn decomposed representations of the underlying factors of data; this becomes significantly more challenging when there are confounding factors (which influence both the cause and the effect). In this paper, we take a generative approach that builds on the recent advances in Variational Auto-Encoders to simultaneously learn those underlying factors as well as the causal effects. We propose a progressive sequence of models, where each improves over the previous one, culminating in the Hybrid model. Our empirical results demonstrate that the performance of the proposed hybrid models are superior to both state-of-the-art discriminative as well as other generative approaches in the literature.

## 1 Introduction

As one of the main tasks in studying causality (Peters et al., 2017; Guo et al., 2018), the goal of Causal Inference is to determine *how much* the value of a certain variable would change (*i.e.*, the **effect**) had another specified variable (*i.e.*, the cause) changed its value. A prominent example is the counterfactual question (Rubin, 1974; Pearl, 2009) "Would this patient have **lived longer** (and by *how much*), had she received an alternative treatment?". Such questions are often asked in the context of precision medicine, — *i.e.*, the customization of health-care tailored to each individual patient — hoping to identify which medical procedure $t \in \mathcal{T}$ will benefit a certain patient $x$ the most, in terms of the treatment outcome $y \in \mathbb{R}$ (*e.g.*, survival time).

$^{W8Yz:}$ A dataset $\mathcal{D} = \{ [x_i, t_i, y_i] \}_{i=1}^{N}$ used for treatment effect estimation has the following format: for the $i^{th}$ instance (*e.g.*, patient), we have some context information $x_i \in \mathcal{X} \subseteq \mathbb{R}^K$ (*e.g.*, age, BMI, blood work, etc.), the administered treatment $t_i$ chosen from a set of treatment options $\mathcal{T}$ (*e.g.*, {0: medication, 1: surgery}), and the associated observed outcome $y_i \in \mathcal{Y}$ (*e.g.*, survival time: $\mathcal{Y} \subseteq \mathbb{R}^+$) as a result of receiving treatment $t_i$.

The first challenge with causal inference is the **unobservability** (Holland, 1986) of the counterfactual outcomes (*i.e.*, outcomes pertaining to the treatments that were *not* administered). In other words, the true causal effect is never observed (*i.e.*, missing in any training data) and cannot be used to train predictive models, nor can it be used to evaluate a proposed model. This makes estimating causal effects a more difficult problem than that of generalization in the supervised learning paradigm. The second common challenge is that the training data is often an observational study that exhibits **selection bias** (Imbens & Rubin, 2015) — *i.e.*, the treatment assignment can depend on the subjects' attributes (as opposed to a Randomized Controlled Trial (RCT) setting). The general problem setup of causal inference and its challenges are described in detail in Appendix A.

Like any other machine learning task, we can employ either of the two general approaches to address the problem of causal inference: (i) discriminative modeling, or (ii) generative modeling, which differ in how the input features $x$ and their target values $y$ are modeled (Ng & Jordan, 2002):

**Discriminative methods** focus solely on modeling the conditional distribution $\Pr(y \,|\, x)$ with the goal of direct prediction of the target $y$ for each instance $x$. For prediction tasks, discriminative approaches are often more accurate since they use the model parameters more efficiently than generative approaches. Most

of the current causal inference approaches are discriminative, including the matching-based methods such as Deep Match (Kallus, 2020) and Counterfactual Propagation (Harada & Kashima, 2020), as well as the regression-based methods such as Balancing Neural Network (BNN) (Johansson et al., 2016), CounterFactual Regression Network (CFR-Net) (Shalit et al., 2017) and its extensions (*cf.*, (Yao et al., 2018; Hassanpour & Greiner, 2019)), Similarity preserved Individual Treatment Effect (SITE) (Yao et al., 2018), and Dragon-Net (Shi et al., 2019).

**Generative methods**, on the other hand, describe the relationship between $x$ and $y$ by their joint probability distribution $\Pr(x, y)$. This, in turn, allows the generative model to answer arbitrary queries, including coping with missing features using the marginal distribution $\Pr(x)$ or [similar to discriminative models] predicting the unknown target values $y$ via $\Pr(y \mid x)$. A promising direction forward for causal inference is developing *generative* models, using either Generative Adverserial Network (GAN) (Goodfellow et al., 2014) or Variational Auto-Encoder (VAE) (Kingma & Welling, 2014; Rezende et al., 2014). This has led to three generative approaches for causal inference: GANs for inference of Individualised Treatment Effects (GANITE) (Yoon et al., 2018), Causal Effect VAE (CEVAE) (Louizos et al., 2017), and Treatment Effect by Disentangled VAE (TEDVAE) (Zhang et al., 2021). However, these generative methods either do not achieve competitive performance compared to the discriminative approaches or come short of fully disentangling the underlying factors of observational data (see Figure 1).

*BLq7 & sCX6:* Our motivation to use a generative model for learning representations of the underlying factors as opposed to a discriminative model is the intuition that learning the underlying data generating process is quite important for causal inference due to the interventional queries that we need to make. That is, knowing how the data was generated would facilitate accurate estimation of what would have happened had a certain variable had taken a different value. Moreover, although discriminative models have excellent predictive performance, they often suffer from two drawbacks: (i) overfitting, and (ii) making highly-confident predictions, even for instances that are "far" from the observed training data. Generative models based on Bayesian inference, on the other hand, can handle both of these drawbacks: issue (i) can be minimized by taking an average over the posterior distribution of model parameters; and issue (ii) can be addressed by explicitly providing model uncertainty via the posterior (Gordon & Hernández-Lobato, 2020). Although the exact inference is often intractable, efficient approximations to the parameter posterior distribution is possible through variational methods. In this work, we use the Variational Auto-Encoder (VAE) framework (Kingma & Welling, 2014; Rezende et al., 2014) to tackle this.

*BLq7 & W8Yz:* In this work, we make the following 3 assumptions that are necessary for the causal effects to be identifiable (*i.e.*, can be uniquely determined from observational data):

**Assumption 1: Stable Unit Treatment Value Assumption (SUTVA).** The potential outcomes for any unit (think "patient") do not vary with the treatments assigned to other units.[c2] Moreover, for each unit, there are no differences in forms or versions of each treatment level, that lead to different potential outcomes.[c2]

**Assumption 2: Unconfoundedness/Ignorability.** There are no *unobserved* confounders — *i.e.*, covariates that contribute to both treatment selection procedure as well as determination of outcomes. $\{Y^t\}_{t \in \mathcal{T}} \perp\!\!\!\perp T \mid X$

**Assumption 3: Overlap.** Every individual $x$ should have a non-zero chance of being assigned to any treatment arm. That is, $0 < \Pr(T = t \mid X = x) < 1 \quad \forall t \in \mathcal{T}, \forall x \in \mathcal{X}$

**Contributions:** We propose three interrelated Bayesian models (namely Series, Parallel, and Hybrid) that employ the VAE framework to address the task of causal inference for binary treatments. We demonstrate that all three of these models significantly outperform the state-of-the-art in terms of estimating treatment effects on two publicly available benchmarks, as well as a fully synthetic dataset that allows for detailed performance analyses. We also show that our proposed Hybrid model is the best at decomposing the underlying factors of any observational dataset.

---

[c2] *I.e.*, patients do not compete to get a certain treatment. Hence, SUTVA does not apply to cases such as organ transplantation, where there is limited supply of organs (so if patient A gets the organ, then patient B does not).

[c2] *I.e.*, patients in each treatment arm get the exact same treatment (*e.g.*, dosage, procedure, etc.).

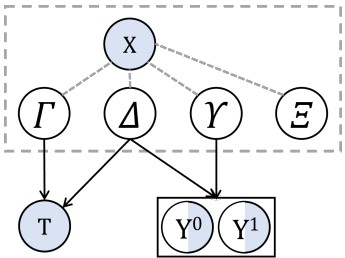

Figure 1: The three non-noise non-overlapping underlying factors of an observational data (namely $\Gamma$, $\Delta$, and $\Upsilon$) (Hassanpour & Greiner, 2020). *sCX6:* $\Gamma$ determines only the treatment, $\Upsilon$ determines only the outcome, and $\Delta$ represents the confounding factors that contribute to both treatment as well as outcome. $\Xi$ represents noise. *BLq7:* $X$ is built as the concatenation of $\Gamma$, $\Delta$, and $\Upsilon$.

## 2 Related Works

For notation, we will use $x$ to describe an instance, and let $t \in \{0, 1\}$ refer to the treatment administered, yielding the outcome with value $y \in \mathbb{R}$; see Appendix A for details.

**CFR-Net** Shalit et al. (2017) considered the binary treatment task and attempted to learn a representation space $\Phi$ that reduces selection bias by making $\Pr(\Phi(x) \mid t = 0)$ and $\Pr(\Phi(x) \mid t = 1)$ as close to each other as possible, provided that $\Phi(x)$ retains enough information that the learned regressors $\{h^t(\Phi(\cdot)) : t \in \{0, 1\}\}$ can generalize well on the observed outcomes. Their objective function includes $L[y_i, h^{t_i}(\Phi(x_i))]$, which is the loss of predicting the observed outcome for instance $i$ (described as $x_i$), weighted by $\omega_i = \frac{t_i}{2u} + \frac{1 - t_i}{2(1-u)}$, where $u = \Pr(t = 1)$. This is effectively setting $\omega_i = \frac{1}{2 \Pr(t_i)}$ where $\Pr(t_i)$ is the probability of selecting treatment $t_i$ over the entire population.

**DR-CFR** Hassanpour & Greiner (2020) argued against the standard implicit assumption that *all* of the covariates $X$ are confounders (*i.e.*, contributing to both treatment assignment and outcome determination). Instead, they proposed the graphical model shown in Figure 1 (with the non-noise underlying factors $\Gamma$, $\Delta$, and $\Upsilon$) and designed a *discriminative* causal inference approach accordingly. Specifically, their "Disentangled Representations for CFR" (DR-CFR) model includes three representation networks, each trained with constraints to ensure that each component corresponds to its respective underlying factor. While the idea behind DR-CFR provides an interesting intuition, it is known that only generative models (and not discriminative ones) can truly identify the underlying data generating mechanism. This paper is a step in this direction.

**Dragon-Net** Shi et al. (2019)'s main objective was to estimate the Average Treatment Effect (ATE), which they explain requires a two stage procedure: (i) fit models that predict the outcomes for each treatment; and (ii) find a downstream estimator of the effect. Their method is based on a classic result from strong ignorability (*i.e.*, Theorem 3 in (Rosenbaum & Rubin, 1983)) that states:

$$(y^1, y^0) \perp\!\!\!\perp t \mid x \qquad \& \qquad \Pr(t = 1 \mid x) \in (0, 1) \qquad \Longrightarrow$$
$$(y^1, y^0) \perp\!\!\!\perp t \mid b(x) \qquad \& \qquad \Pr(t = 1 \mid b(x)) \in (0, 1)$$

where $b(x)$ is a balancing score[c0] (here, propensity) and argued that only the parts of $X$ relevant for predicting $T$ are required for the estimation of the causal effect.[c0] This theorem only provides a way to *match* treated and control instances though — *i.e.*, it helps finding potential counterfactuals from the alternative group, which they use to calculate ATE. Shi et al. (2019), however, used this theorem to derive minimal representations on which to *regress* to estimate the outcomes.

**GANITE** Yoon et al. (2018) proposed the counterfactual GAN, whose generator $G$, given $\{x, t, y^t\}$, estimates the counterfactual outcomes ($\hat{y}^{\neg t}$); and whose discriminator $D$ tries to identify which of

---

[c0]That is, $X \perp\!\!\!\perp T \mid b(X)$ (Rosenbaum & Rubin, 1983).

[c0]The authors acknowledge that this would hurt the predictive performance for individual outcomes. As a result, this yields inaccurate estimation of Individual Treatment Effects (ITEs).

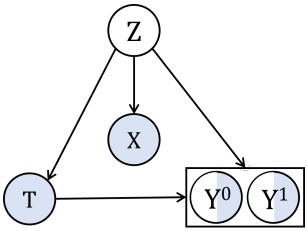

Figure 2: Graphical model of the CEVAE method (Louizos et al., 2017)

$\{[x, 0, y^0], [x, 1, y^1]\}$ is the factual outcome. It is, however, unclear why this requires that $G$ must produce samples that are indistinguishable from the factual outcomes, especially as $D$ can just learn the *treatment selection mechanism* (*i.e.*, the mapping from $X$ to $T$) instead of distinguishing the factual outcomes from counterfactuals. Although this work is among the few generative approaches for causal inference, our empirical results (in Section 4) show that it does not effectively estimate counterfactual outcomes.

**CEVAE**    Louizos et al. (2017) used VAE to extract latent confounders from their observed proxies in $X$. While this is a step in the right direction, empirical results show that it does not always accurately estimate treatment effects (see Section 4). The authors note that this may be because CEVAE is not able to address the problem of selection bias. Another reason for CEVAE's sub-optimal performance might be its assumed graphical model of the underlying data generating mechanism, depicted in Figure 2. This model assumes that there is only one latent variable $Z$ (confounding $T$ and $Y$) that generates the entire observational data; however, (Kuang et al., 2017; Hassanpour & Greiner, 2020) have shown the advantages of involving more factors (see Figure 1).

**TEDVAE**    Similar to DR-CFR (Hassanpour & Greiner, 2020), Zhang et al. (2021) proposed TEDVAE in an attempt to learn disentangled factors but using a *generative* model instead (*i.e.*, a VAE with a three-headed encoder, one for each underlying factor). While their method proposed an interesting intuition on how to achieve this task, according to the reported empirical results (see their Figure 4c), the authors found that TEDVAE is not successful in identifying the risk factors $z_y$ (equivalent to our $\Upsilon$). This might be because their model does not have a mechanism for distinguishing between the risk factors and confoundings $z_c$ (equivalent to our $\Delta$). The evidence is in their Equation (8), which would allow $z_y$ to be degenerate and have all information embedded in $z_c$. Our work, however, proposes an architecture that can achieve this decomposition.

**M1 and M2 VAEs**    The M1 model (Kingma & Welling, 2014) is the conventional VAE, which learns a latent representation from the covariate matrix $X$ alone in an unsupervised manner. Kingma et al. (2014) extended this to the M2 model that, in addition to the covariates, also allows the target information to guide the representation learning process in a semi-supervised manner. Stacking the M1 and M2 models produced their best results: first learn a representation $Z_1$ from the raw covariates, then find a second representation $Z_2$, now learning from $Z_1$ (instead of the raw data) as well as the target information. Appendix B.1 presents a more detailed overview of the M1 and M2 VAEs. In our work, the target information includes the treatment bit $T$ as well as the observed outcome $Y$.[c0] This additional information helps the model to learn more expressive representations, which was not possible with the unsupervised M1 model.

## 3   Method

Following (Kuang et al., 2017; Hassanpour & Greiner, 2020) and without loss of generality, we assume that $X$ follows an unknown joint probability distribution $\Pr(X \mid \Gamma, \Delta, \Upsilon, \Xi)$, where $\Gamma$, $\Delta$, $\Upsilon$, and $\Xi$ are non-overlapping independent factors. Moreover, we assume that the treatment $T$ follows $\Pr(T \mid \Gamma, \Delta)$ (*i.e.*, $\Gamma$ and $\Delta$ are responsible for selection bias) and the outcome $Y^T$ follows $\Pr_T(Y^T \mid \Delta, \Upsilon)$ — see Figure 1. Observe that the factor $\Gamma$ (respectively, $\Upsilon$) partially determines only $T$ (respectively, $Y$), but not $Y$ (respectively, $T$); and $\Delta$ includes the confounding factors between $T$ and $Y$.

---

[c0]Therefore, we require multiple stacked models here.

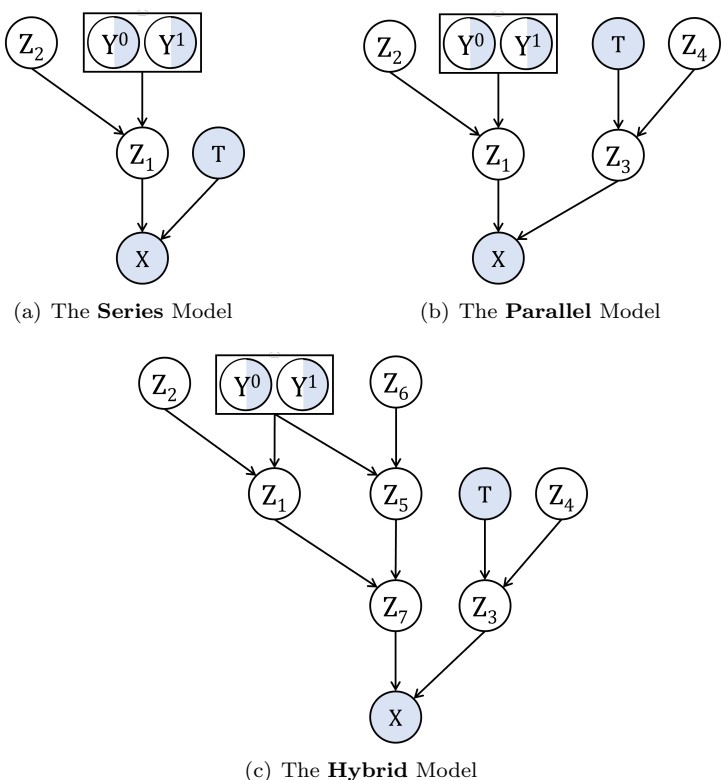

(a) The **Series** Model        (b) The **Parallel** Model

(c) The **Hybrid** Model

Figure 3: Belief nets of the proposed models.

We emphasize that the belief net in Figure 1 is built without loss of generality; *i.e.*, it also covers the scenarios where any of the latent factors is degenerate *BLq7:* *(i.e., has a zero-dimensionality; effectively making it non-existent in $X$)*. Therefore, if we can design a method that has the capacity to capture all of these latent factors, it would be successful in all scenarios — even in the ones that have degenerate factors (and in fact this is true; see the experimental setting and results of the Synthetic benchmark in Sections 4.1 and 4.3 respectively).

Our goal is to design a generative model architecture that encourages learning decomposed representations of these underlying latent factors (see Figure 1). In other words, it should be able to decompose and separately learn the three underlying factors that are responsible for determining "$T$ only" ($\Gamma$), "$Y$ only" ($\Upsilon$), and "both $T$ and $Y$" ($\Delta$). To achieve this, we propose a progressive sequence of three models (namely Series, Parallel, and Hybrid; as illustrated in Figures 3(a), 3(b), and 3(c) respectively), where each is an *improvement* over the previous one. Every model employs several stacked M2 or M1+M2 VAEs (Kingma et al., 2014). Each of these components includes a decoder (generative model) and an encoder (variational posterior), which are parametrized as deep neural networks. *sCX6 & W8Yz:* Note that the M2 and M1+M2 components in the proposed architectures are capable of employing the extra supervision from $Y$ and $T$ to help learning disentangled representations of the underlying factors. In other words, in the proposed architectures, $Y$ and $T$ guide each of the representation nodes to capture the appropriate factor.

## 3.1 The Variational Auto-Encoder Component

### 3.1.1 The Series Model

The belief net of the Series model is illustrated in Figure 3(a). Louizos et al. (2015) proposed a similar architecture to address fairness in machine learning, but using a binary sensitive variable $S$ (*e.g.*, gender, race, etc.) rather than the treatment $T$. Here, we employ this architecture for causal inference and explain

why it should work. We hypothesize that this structure functions as a fractionating column[c1]: the bottom M2 VAE attempts to decompose $\Gamma$ (guided by $T$) from $\Delta$ and $\Upsilon$ (captured by $Z_1$); and the top M2 VAE attempts to learn $\Delta$ and $\Upsilon$ (guided by $Y$).

The decoder and encoder components of the Series model — $p(\cdot)$ and $q(\cdot)$ parametrized by $\theta_s$ and $\phi_s$ respectively — involve the following distributions:

| Priors | | Likelihood | Posteriors | |
|---|---|---|---|---|
| $p_{\theta_s}(z_2)$ | $p_{\theta_s}(z_1\|y,z_2)$ | $p_{\theta_s}(x\|z_1,t)$ | $q_{\phi_s}(z_1\|x,t)$ | $q_{\phi_s}(y\|z_1)$ |
| | | | $q_{\phi_s}(z_2\|y,z_1)$ | |

Hereafter, we drop the $\theta$ and $\phi$ subscripts for brevity.

The goal is to maximize the conditional log-likelihood of the observed data[c0] (left-hand-side of the following inequality) by maximizing the Evidence Lower BOund (ELBO; right-hand-side) — *i.e.*,

$$\sum_{i=1}^{N} \log p(x_i|t_i, y_i) \quad \geq \quad \sum_{i=1}^{N} \mathbb{E}_{q(z_1|x,t)}\big[\log p(x_i|z_{1_i}, t_i)\big] \tag{1}$$

$$-\mathrm{KL}\big(q(z_1|x,t) \,||\, p(z_1|y,z_2)\big) - \mathrm{KL}\big(q(z_2|y,z_1) \,||\, p(z_2)\big) \tag{2}$$

where KL denotes the Kullback-Leibler divergence, $p(z_2)$ is the unit multivariate Gaussian (*i.e.*, $\mathcal{N}(0,\mathbb{I})$), and the other distributions are [BLq7:]assumed to be multivariate Gaussian whose $\mu$ and $\Sigma$ (diagonal) are parameterized as deep neural networks.

### 3.1.2 The Parallel Model

The Series model is composed of two M2 stacked models. However, Kingma et al. (2014) showed that an M1+M2 stacked architecture learns better representations than an M2 model alone for a downstream prediction task. This motivated us to design a double M1+M2 Parallel model; where one arm is for the outcome to guide the representation learning via $Z_1$ and another for the treatment to guide the representation learning via $Z_3$. Figure 3(b) shows the belief net of this model. We hypothesize that $Z_1$ would learn $\Delta$ and $\Upsilon$, and $Z_3$ would learn $\Gamma$ (and perhaps partially $\Delta$).

The decoder and encoder components of the Parallel model — $p(\cdot)$ and $q(\cdot)$ parametrized by $\theta_p$ and $\phi_p$ respectively — involve the following distributions:

| Priors | | Likelihood | Posteriors | |
|---|---|---|---|---|
| $p(z_2)$ | $p(z_1\|y,z_2)$ | $p(x\|z_1,z_3)$ | $q(z_1\|x,t)$ | $q(y\|z_1)$ |
| $p(z_4)$ | $p(z_3\|t,z_4)$ | | $q(z_2\|y,z_1)$ | $q(t\|z_3)$ |
| | | | $q(z_3\|x,y)$ | |
| | | | $q(z_4\|t,z_3)$ | |

Here, the conditional log-likelihood can be upper bounded by:

$$\sum_{i=1}^{N} \log p(x_i|t_i, y_i) \quad \geq \quad \sum_{i=1}^{N} \mathbb{E}_{q(z_1,z_3|x,t,y)}\big[\log p(x_i|z_{1_i}, z_{3_i})\big] \tag{3}$$

$$-\mathrm{KL}\big(q(z_1|x,t) \,||\, p(z_1|y,z_2)\big) - \mathrm{KL}\big(q(z_2|y,z_1) \,||\, p(z_2)\big)$$

$$-\mathrm{KL}\big(q(z_3|x,y) \,||\, p(z_3|t,z_4)\big) - \mathrm{KL}\big(q(z_4|t,z_3) \,||\, p(z_4)\big) \tag{4}$$

---

[c1]In chemistry, a fractionating column is used for separating different liquid compounds in a mixture; see `https://en.wikipedia.org/wiki/Fractionating_column` for more details. In our work, similarly, we can separate different factors from the pool of features using the proposed architectures (especially, the Hybrid model).

[c0 BLq7:] We try to maximize the lower bound on the likelihood of the observed data (*i.e.*, $P(X|Y,T)$), in order to find the underlying data generating process from which our data is sampled. This is the generative part of the algorithm. After learning the representations of the underlying factors, we then use a discriminative approach to estimate the outcomes.

### 3.1.3 The Hybrid Model

The final model, Hybrid (see Figure 3(c)), attempts to combine the best capabilities of the previous two architectures. The backbone of the Hybrid model has a Series architecture, that separates $\Gamma$ (factors related to the treatment $T$; captured by the right module with $Z_3$ as its head) from $\Delta$ and $\Upsilon$ (factors related to the outcome $Y$; captured by the left module with $Z_7$ as its head). The left module, itself, consists of a Parallel model that attempts to proceed one step further and decompose $\Delta$ from $\Upsilon$. This is done with the help of a discrepancy penalty (see Section 3.3).

The decoder and encoder components of the Hybrid model — $p(\cdot)$ and $q(\cdot)$ parametrized by $\theta_h$ and $\phi_h$ respectively — involve the following distributions:

| Priors | | Likelihood | Posteriors | |
|---|---|---|---|---|
| $p(z_2)$ | $p(z_1|y,z_2)$ | $p(x|z_3,z_7)$ | $q(z_1|z_7)$ | $q(y|z_1,z_5)$ |
| $p(z_4)$ | $p(z_3|t,z_4)$ | | $q(z_2|y,z_1)$ | $q(t|z_3)$ |
| $p(z_6)$ | $p(z_5|y,z_6)$ | | $q(z_3|x,y)$ | |
| | $p(z_7|z_1,z_5)$ | | $q(z_4|t,z_3)$ | |
| | | | $q(z_5|z_7)$ | |
| | | | $q(z_6|y,z_5)$ | |
| | | | $q(z_7|x,t)$ | |

Here, the conditional log-likelihood can be upper bounded by:

$$\sum_{i=1}^{N} \log p(x_i|t_i, y_i) \quad \geq \quad \sum_{i=1}^{N} \mathbb{E}_{q(z_3, z_7|x,t,y)}\big[\log p(x_i|z_{3_i}, z_{7_i})\big] \tag{5}$$

$$-\mathrm{KL}\big(q(z_1|z_7) \,\|\, p(z_1|y,z_2)\big) - \mathrm{KL}\big(q(z_2|y,z_1) \,\|\, p(z_2)\big)$$

$$-\mathrm{KL}\big(q(z_3|x,y) \,\|\, p(z_3|t,z_4)\big) - \mathrm{KL}\big(q(z_4|t,z_3) \,\|\, p(z_4)\big)$$

$$-\mathrm{KL}\big(q(z_5|z_7) \,\|\, p(z_5|y,z_6)\big) - \mathrm{KL}\big(q(z_6|y,z_5) \,\|\, p(z_6)\big)$$

$$-\mathrm{KL}\big(q(z_7|x,t) \,\|\, p(z_7|z_1,z_5)\big) \tag{6}$$

For all three of these models, we refer to the first term in the ELBO (*i.e.*, right-hand-side of Equations (1), (3), or (5)) as the Reconstruction Loss (RecL) and the next term(s) (*i.e.*, Equations (2), (4), or (6)) as the KL Divergence (KLD). Concisely, the ELBO can be viewed as maximizing: $\mathrm{RecL} - \mathrm{KLD}$.

## 3.2 Further Disentanglement with $\beta$-VAE

As mentioned earlier, we want the learned latent variables to be disentangled, to match our assumption of non-overlapping factors $\Gamma$, $\Delta$, and $\Upsilon$. To further encourage this (in addition to the proposed architecture), we employ the $\beta$-VAE (Higgins et al., 2017), which adds a hyperparameter $\beta$ as a multiplier of the KLD part of the ELBO. This adjustable hyperparameter facilitates a trade-off that helps balance the latent channel capacity and independence constraints with the reconstruction accuracy. In other words, a higher $\beta$ coefficient for the KL divergence term would enforce the approximate posterior to be closer to the unit Gaussian prior, which in turn would encourage independence between the learned latent variables.[c0] We therefore hypothesize that including the $\beta$ hyperarameter should grant a better control over the level of disentanglement in the learned representations Burgess et al. (2018). Therefore, the generative objective to be minimized becomes:

$$\mathcal{L}_{\mathrm{VAE}} \quad = \quad -\mathrm{RecL} \, + \, \beta \cdot \mathrm{KLD} \tag{7}$$

Although Higgins et al. (2017) suggest setting $\beta$ greater than 1 in most applications, Hoffman et al. (2017) show that having a $\beta < 1$ weight on the KLD term can be interpreted as optimizing the ELBO under an alternative prior, which functions as a regularization term to reduce the chance of degeneracy.

---

[c0]This is due to the zero non-diagonal terms in the covariance matrix of the unit Gaussian prior.

### 3.3 Discrepancy

Although all three proposed models encourage statistical independence between $T$ and $Z_1$ in the marginal posterior $q_\phi(Z_1|T)$ where $X$ is not given — see the collider structure (at $X$): $T \to X \leftarrow Z_1$ in Figure 3(a) — an information leak is quite possible due to the correlation between the outcome $Y$ and treatment $T$ in the data.[c0] We therefore require an extra regularization term on $q_\phi(Z_1|T)$ in order to penalize the discrepancy (denoted by disc) between the conditional distributions of $Z_1$ given $T=0$ versus given $T=1$.[c1] To achieve this regularization, we calculate the disc using an Integral Probability Metric (IPM) (Mansour et al., 2009)[c1] (*cf.*, (Louizos et al., 2015; Shalit et al., 2017; Yao et al., 2018), etc.) that measures the distance between the two above-mentioned distributions:

$$\mathcal{L}_{\texttt{disc}} \quad = \quad \text{IPM}\big( \{z_1\}_{i:t_i=0}, \{z_1\}_{i:t_i=1} \big) \tag{8}$$

### 3.4 Predictive Loss

Note, however, that neither the VAE nor the disc losses contribute to training a predictive model for outcomes. To remedy this, we extend the objective function to include a discriminative term for the regression loss of predicting $y$: [c0]

$$\mathcal{L}_{\text{pred}} \quad = \quad \frac{1}{N} \sum_{i=1}^{N} \omega_i \cdot \mathcal{L}\big[ y_i, \hat{y}_i \big] \tag{9}$$

where the predicted outcome $\hat{y}_i$ [BLq7:]is set to be the mean of the $q_\phi^{t_i}(y_i|z_{1_i})$ distribution for the Series and Parallel models and the mean of the $q_\phi^{t_i}(y_i|z_{1_i}, z_{5_i})$ distribution for the Hybrid model; $\mathcal{L}\big[ y_i, \hat{y}_i \big]$ is the factual loss (*i.e.*, L2 loss for real-valued outcomes and log loss for binary-valued outcomes); and $\omega_i$ represent the weights that attempt to account for selection bias. We consider two approaches in the literature to derive the weights: (i) the *Population-Based* (PB) weights as proposed in (Shalit et al., 2017); and (ii) the *Context-Aware* (CA) weights as proposed in (Hassanpour & Greiner, 2019)[c1]. Note that disentangling $\Delta$ from $\Upsilon$ is only beneficial when using the CA weights, since we need just the $\Delta$ factors to derive them (Hassanpour & Greiner, 2020).

### 3.5 Final Model(s)

Putting everything together, the overall objective function to be minimized is:

$$\mathcal{J} \quad = \quad \mathcal{L}_{\text{pred}} + \alpha \cdot \mathcal{L}_{\texttt{disc}} + \gamma \cdot \mathcal{L}_{\text{VAE}} + \lambda \cdot \mathfrak{Reg} \tag{10}$$

where $\mathfrak{Reg}$ penalizes the model complexity.

This objective function is motivated by the work of (McCallum et al., 2006), which suggested optimizing a convex combination of discriminative and generative losses would indeed improve predictive performance. As an empirical verification, note that for $\gamma=0$, the Series and Parallel models effectively reduce to CFR-Net. However, our empirical results (see Section 4) suggest that the generative term in the objective function helps learning representations that embed more relevant information for estimating outcomes than that of $\Phi$ in CFR-Net.

We refer to the family of our proposed methods as VAE-CI (Variational Auto-Encoder for Causal Inference); specifically: **{S, P, H}-VAE-CI**, for **S**eries, **P**arallel, and **H**ybrid respectively. We anticipate that each method is an *improvement* over the previous one in terms of estimating causal effects, culminating in

---

[c0 BLq7:] In other words, since $Y$ is one of $Z_1$'s parents, and $Y$ is either the outcome of treatment 1 or 0, we have an information leak from $T$ to $Z_1$ through $Y$.

[c1]Note that even for the Hybrid model (see Figure 3(c)), we apply the disc penalty only on $Z_1$ and not $Z_7$. This is because we want $Z_1$ to capture $\Upsilon$ and $Z_5$ to capture $\Delta$ (so $Z_5$ should have a non-zero disc). Hence, $Z_7$ must include both $\Delta$ and $\Upsilon$ (and therefore, it should have a non-zero disc) to be able to reconstruct $X$.

[c1]In this work, we use the Maximum Mean Discrepancy (MMD) (Gretton et al., 2012) as our IPM.

[c0]This is similar to the way Kingma et al. (2014) included a classification loss in their Equation (9).

[c1 BLq7:] Note that we use $Z_1$ (which captures both $\Delta$ and $\Upsilon$) for deriving the weights for the series and parallel models and $Z_5$ (which captures $\Delta$) for the hybrid model.

**H-VAE-CI**, which we expect can best decompose the underlying factors and accurately estimate the outcomes of all treatments.

## 4 Experiments, Results, and Discussion

### 4.1 Benchmarks

Researchers often evaluate treatment effect estimation on semi- or fully- synthetic datasets that include both factual and counterfactual outcomes.

1. **Infant Health and Development Program (IHDP)** The original IHDP randomized controlled trial was designed to evaluate the effect of specialist home visits on future cognitive test scores of premature infants. Hill (Hill, 2011) induced selection bias by removing a non-random subset of the treated population. The dataset contains 747 instances (608 control and 139 treated) with 25 covariates. We worked with the same dataset provided by and used in Shalit et al. (2017); Johansson et al. (2016; 2018) $^{sCX6:}$ , in which outcomes are simulated as setting "A" of the Non-Parametric Causal Inference (NPCI) package Dorie (2016). The noiseless outcomes are used to compute the true individual effects (available for evaluation purpose only).

2. **Atlantic Causal Inference Conference 2018 (ACIC'18)** ACIC'18 is a collection of binary-treatment observational datasets released for a data challenge. $^{sCX6:}$ Each dataset is created according to a unique data generating process (unknown to the challenge participants) that describe the relationship between the treatment assignment, the outcomes, and the covariates. For the evaluation purposes, the organizers had not only supplied the factual outcomes but also the counterfactual outcomes as well. Following (Shi et al., 2019), we used those datasets with $N \in \{1, 5, 10\} \times 10^3$ instances (four datasets in each category). The covariates matrix for each dataset involves 177 features and is sub-sampled from a table of medical measurements taken from the Linked Birth and Infant Death Data (LBIDD) (MacDorman & Atkinson, 1998), that contains information corresponding to 100,000 subjects.

3. **Fully Synthetic Datasets** We generated a set of synthetic datasets according to the procedure described in (Hassanpour & Greiner, 2020) (details in Appendix C.1). We considered all the viable datasets in a mesh generated by various sets of variables, of sizes $m_\Gamma, m_\Delta, m_\Upsilon \in \{0, 4, 8\}$ and $m_\Xi = 1$. This creates 24 scenarios[c2] that consider all relevant combinations of sizes of $\Gamma$, $\Delta$, and $\Upsilon$ (corresponding to different levels of selection bias). For each scenario, we synthesized multiple datasets with various initial random seeds to allow for statistical significance testing of the performance comparisons between the contending methods.

### 4.2 Identification of the Underlying Factors

#### 4.2.1 Procedure for Evaluating Identification of the Underlying Factors

To evaluate $^{BLq7\ \&\ sCX6:}$ any representation network $Z_j$ in terms of its disentanglement quality of the learned representations of the underlying factors, we use a fully synthetic dataset (#3 above) with $m_\Gamma = m_\Delta = m_\Upsilon = 8$ and $m_\Xi = 1$. We then ran the learned model on four dummy test instances $V_i \in \mathbb{R}^{m_\Gamma + m_\Delta + m_\Upsilon + m_\Xi}$ as depicted on the left-side of Figure 4. The first to third vectors had "1" (constant) in the positions associated with $\Gamma$, $\Delta$, and $\Upsilon$ respectively, and the remaining $2 \times 8 + 1 = 17$ positions were filled with "0". The fourth vector was all "1" except for the last position (the noise) which was "0". $^{BLq7\ \&\ sCX6:}$The use of these dummy signals as input helps measure the maximum amount of information that is $^{BLq7\ \&\ sCX6:}$ allowed to reach the final layer of each representation network.

Next, each vector $V_i$ is fed $^{BLq7\ \&\ sCX6:}$as input to each trained network $Z_j$. We let $O_{i,j}$ be the output (here, $\in \mathbb{R}^{200}$ $^{BLq7\ \&\ sCX6:}$with an activation function of `elu` ) of the encoder network $Z_j$ when its input is set to $V_i$.

---

[c2]There are $3^3 = 27$ combinations in total; however, we removed three of them that generate pure noise outcomes — *i.e.*, $\Delta = \Upsilon = \emptyset$: $(0, 0, 0)$, $(4, 0, 0)$, and $(8, 0, 0)$.

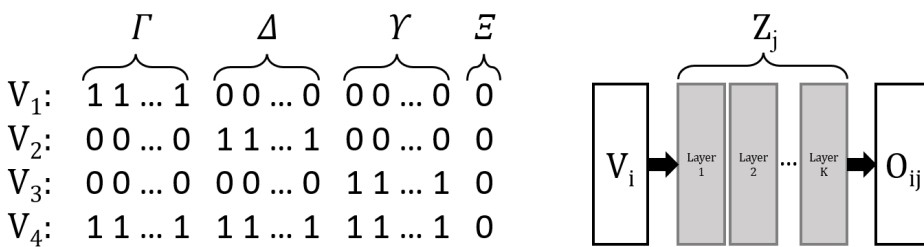

Figure 4: The four dummy $x$-like vectors (left); and the input/output vectors of the representation networks (right).

| H-VAE-CI | | | | | | | |
|---|---|---|---|---|---|---|---|
| | Z1 | Z2 | Z3 | Z4 | Z5 | Z6 | Z7 |
| Γ | 0.2262 | 0.2599 | **1.2512** | 1.1979 | 0.3109 | 0.2890 | 0.2959 |
| Δ | 0.6431 | 0.7704 | 0.5202 | 0.5602 | **0.8229** | 0.7073 | **0.7170** |
| Υ | **0.8957** | 0.7929 | 0.2622 | 0.2641 | 0.6551 | 0.5679 | **0.7106** |

| S-VAE-CI | | |
|---|---|---|
| | Z1 | Z2 |
| Γ | 0.3407 | 0.3451 |
| Δ | **0.7925** | 0.7828 |
| Υ | **0.8804** | 0.8204 |

| P-VAE-CI | | | |
|---|---|---|---|
| | Z1 | Z2 | Z3 | Z4 |
| Γ | 0.2920 | 0.2922 | **0.6374** | 0.6676 |
| Δ | **0.7799** | 0.7563 | 0.1882 | 0.1888 |
| Υ | **0.7686** | 0.7735 | 0.4738 | 0.3737 |

| DR-CFR | | | |
|---|---|---|---|
| | Γ_rep | Δ_rep | Υ_rep |
| Γ | **0.5289** | 0.1733 | 0.8412 |
| Δ | **0.5429** | **0.7038** | 0.8402 |
| Υ | 0.0544 | **0.6749** | 1.0008 |

Figure 5: Performance analysis for decomposition of the underlying factors on the **Synthetic** dataset with $m_\Gamma = m_\Delta = m_\Upsilon = 8$ and $m_\Xi = 1$. The color shading in each cell represents the value of that cell, with a longer colored bar for larger values.

The average of the 200 values of the $O_{i,j}$ (*i.e.*, $Avg(O_{i,j})$) represents the power of signal that was produced by the $Z_j$ channel on the input $V_i$.[c7] The values reported in the tables illustrated in Figure 5 are the ratios of $Avg(O_{1,j})$, $Avg(O_{2,j})$, and $Avg(O_{3,j})$ divided by $Avg(O_{4,j})$ for each of the learned representation networks. Note that, a larger ratio indicates that the respective representation network $Z_j$ has allowed more of the input signal $V_i$ to pass through; *BLq7 & sCX6*: thus, $Z_j$ has in fact captured the respective underlying factor. [c9]

If the model could perfectly learn each underlying factor in a disentangled manner, we expect to see one element in each column to be significantly larger than the other elements in that column. For example, for H-VAE-CI, the weights of the $\Gamma$ row should be highest for $Z_3$, as that means $Z_3$ has captured the $\Gamma$ factors. Similarly, we would want the $Z_5$ and $Z_1$ entries on the $\Delta$ and $\Upsilon$ rows to be largest respectively.

### 4.2.2 Results' Analysis

As expected, Figure 5 shows that $Z_3$ and $Z_4$ capture $\Gamma$ (*e.g.*, the $Z_3$ ratios for $\Gamma$ in the {P, H}-VAE-CI tables are largest), and $Z_1$, $Z_2$, $Z_5$, $Z_6$, and $Z_7$ capture $\Delta$ and $\Upsilon$. Note that decomposition of $\Delta$ from $\Upsilon$ has not been achieved by any of the methods except for H-VAE-CI, which captures $\Upsilon$ by $Z_1$ and $\Delta$ by $Z_5$ (note the ratios are largest for $Z_1$ and $Z_5$). This decomposition is vital for deriving context-aware importance sampling weights because they must be calculated from $\Delta$ only (Hassanpour & Greiner, 2020). Also observe that {P, H}-VAE-CI are each able to separate $\Gamma$ from $\Delta$. However, DR-CFR, which tried to disentangle all factors, failed not only to disentangle $\Delta$ from $\Upsilon$, but also $\Gamma$ from $\Delta$.

---

[c7] *BLq7 & sCX6*: *E.g.*, if a representation network $Z_j$ is supposed to, for example, capture $\Delta$, we expect to see the output of $Z_j$ network to be high for input of $V_2$ and low for inputs of $V_1$ and $V_3$ dummy variables.

[c9] Unlike the evaluation strategy presented in (Hassanpour & Greiner, 2020) that only examined the first layer's weights of each representation network, we propagate the values through the entire network and check how much of each factor is exhibited in the final layer of every representation network.

### 4.3 Treatment Effect Estimation

There are two categories of performance measures:

**Individual-based:** "Precision in Estimation of Heterogeneous Effect" (Hill, 2011): PEHE $= \sqrt{\frac{1}{N}\sum_{i=1}^{N}(\hat{e}_i - e_i)^2}$ uses $\hat{e}_i = \hat{y}_i^1 - \hat{y}_i^0$ as the estimated effect and $e_i = y_i^1 - y_i^0$ as the true effect.

**Population-based:** "Bias of the Average Treatment Effect": $\epsilon_{\text{ATE}} = \left| \text{ATE} - \widehat{\text{ATE}} \right|$ where $\text{ATE} = \frac{1}{N}\sum_{i=1}^{N} y_i^1 - \frac{1}{N}\sum_{j=1}^{N} y_j^0$ and $\widehat{\text{ATE}}$ follows the same formula except that it is calculated based on the estimated outcomes.

In this paper, we compare performances of the proposed **{S, P, H}-VAE-CI** versus the following treatment effect estimation methods: **CFR-Net** (Shalit et al., 2017), **DR-CFR** (Hassanpour & Greiner, 2020), **Dragon-Net** (Shi et al., 2019), **GANITE** (Yoon et al., 2018), **CEVAE** (Louizos et al., 2017), and **TEDVAE** (Zhang et al., 2021). The basic search grid for hyperparameters of the CFR-Net based algorithms (including our methods) is available in Appendix C.2. To ensure that our performance gain is not merely based on an increased complexity of the models, we also performed our grid search for all the contending methods with an updated number of layers and/or number of neurons in each layer, This guaranteed that all methods have the same number of parameters and therefore a similar model complexity.

We ran the experiments for the contender methods using their publicly available code-bases; note the following points regarding these runs:

- Since Dragon-Net is designed to estimate ATE only, we did not report its performance results for the PEHE measure (which, as expected, were significantly inaccurate).

- Original GANITE code-base could only deal with binary outcomes. We modified the code (losses, etc.) to allow it to process real-valued outcomes also.

- We were surprised that CEVAE diverged when running on the ACIC'18 datasets. To avoid this, we had to run the ACIC'18 experiments on the binary covariates only.

#### 4.3.1 Results' Analysis

Table 1 summarizes the mean and standard deviation of the PEHE and $\epsilon_{\text{ATE}}$ measures (lower is better) on the IHDP, ACIC'18, and Synthetic benchmarks. VAE-CI achieves the best performance among the contending methods. These results are statistically significant (in **bold**; based on the Welch's unpaired t-test with $\alpha = 0.05$) for the IHDP and Synthetic benchmarks. Although VAE-CI also achieves the best performance on the ACIC'18 benchmark, the results are not statistically significant due to the high standard deviation of the performances of the contending methods.

Figure 6 visualizes the PEHE measures on the entire synthetic datasets with sample size of $N = 10{,}000$. We observe that both plots corresponding to H-VAE-CI method (PB as well as CA) are completely within the plots of all other methods, showcasing H-VAE-CI's superior performance under every possible selection bias scenario. Note that for scenarios where $m_\Delta = 0$ (*i.e.*, the ones of the form $m_\Gamma\_0\_m_\Upsilon$ on the perimeter of the radar chart in Figure 6), the performances of H-VAE-CI (PB) and H-VAE-CI (CA) are almost identical. This is expected, since for these scenarios, the learned representation for $\Delta$ would be degenerate, and therefore, the context-aware weights would reduce to population-based ones. On the other hand, for scenarios where $m_\Delta \neq 0$, the H-VAE-CI (CA) often performs better than H-VAE-CI (PB). This may be because H-VAE-CI has correctly disentangled $\Delta$ from $\Upsilon$. This facilitates learning good CA weights that better account for selection bias, which in turn, results in a better causal effect estimation performance.

### 4.4 Hyperparameters' Sensitivity Analyses

Figure 7 illustrates the results of our hyperparameters' sensitivity analyses (in terms of PEHE). In the following, we discuss the insights we gained from these ablation studies:

Table 1: PEHE and $\epsilon_{\text{ATE}}$ performance measures (lower is better) represented in the form of "mean (standard deviation)". The **bold** values in each column are the best (statistically significant based on the Welch's unpaired t-test with $\alpha = 0.05$).

| Method | IHDP | | ACIC'18 | | Synthetic | |
|---|---|---|---|---|---|---|
| | **PEHE** | $\epsilon_{\text{ATE}}$ | **PEHE** | $\epsilon_{\text{ATE}}$ | **PEHE** | $\epsilon_{\text{ATE}}$ |
| **CFR-Net** | 0.75 (0.57) | 0.08 (0.10) | 5.13 (5.59) | 1.21 (1.81) | 0.39 (0.08) | 0.027 (0.020) |
| **DR-CFR** | 0.65 (0.37) | 0.03 (0.04) | 3.86 (3.39) | 0.80 (1.41) | 0.26 (0.07) | 0.007 (0.004) |
| **Dragon-Net** | NA | 0.14 (0.15) | NA | 0.48 (0.77) | NA | 0.007 (0.005) |
| **GANITE** | 2.81 (2.30) | 0.24 (0.46) | 3.55 (2.27) | 0.69 (0.65) | 1.28 (0.43) | 0.036 (0.015) |
| **CEVAE** | 2.50 (3.47) | 0.18 (0.25) | 5.30 (5.52) | 3.29 (3.50) | 1.39 (0.32) | 0.287 (0.217) |
| **TEDVAE** | 1.61 (2.37) | 0.18 (0.23) | 6.63 (8.69) | 3.74 (5.00) | 0.25 (0.07) | 0.013 (0.007) |
| **S-VAE-CI** | **0.51 (0.37)** | **0.00 (0.02)** | 2.73 (2.39) | 0.51 (0.82) | 0.28 (0.05) | 0.004 (0.003) |
| **P-VAE-CI** | **0.52 (0.36)** | **0.01 (0.03)** | 2.62 (2.26) | 0.37 (0.75) | 0.28 (0.05) | 0.004 (0.003) |
| **H-VAE-CI (PB)** | **0.49 (0.36)** | **0.01 (0.02)** | 1.78 (1.27) | 0.44 (0.77) | **0.20 (0.03)** | **0.003 (0.002)** |
| **H-VAE-CI (CA)** | **0.48 (0.35)** | **0.01 (0.01)** | 1.66 (1.30) | 0.39 (0.75) | **0.18 (0.02)** | **0.003 (0.002)** |

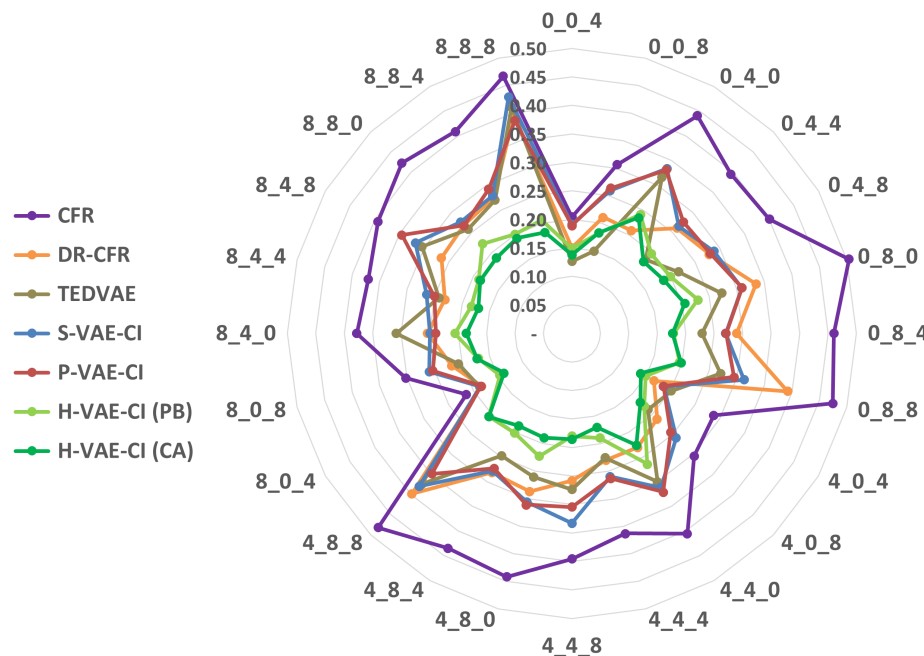

Figure 6: Radar graphs of PEHE (on the radii; closer to the center is better) for the entire **Synthetic** benchmark ($24 \times 3$ with $N = 10{,}000$; each vertex denotes the respective dataset). Figure is best viewed in color.

- For the $\alpha$ hyperparameter (*i.e.*, coefficient of the discrepancy penalty), Figure 7(a) suggests that DR-CFR and H-VAE-CI methods have the most robust performance throughout various values of $\alpha$. This is expected, because, unlike CFR-Net and {S, P}-VAE-CI, DR-CFR and H-VAE-CI possess an independent node for representing $\Delta$. This helps them still capture $\Delta$ as $\alpha$ grows; since for them, $\alpha$ only affects learning a representation of $\Upsilon$. Comparing H-VAE-CI (PB) with (CA), we observe that for all $\alpha > 0.01$, (CA) outperforms (PB). This is because the discrepancy penalty would force $Z_1$ to only capture $\Upsilon$ and $Z_5$ to only capture $\Delta$. This results in deriving better CA weights (that should be learned from $\Delta$; here, from its learned representation $Z_5$). H-VAE-CI (PB), on the other hand, cannot take advantage of this disentanglement, which explains its sub-optimal performance.

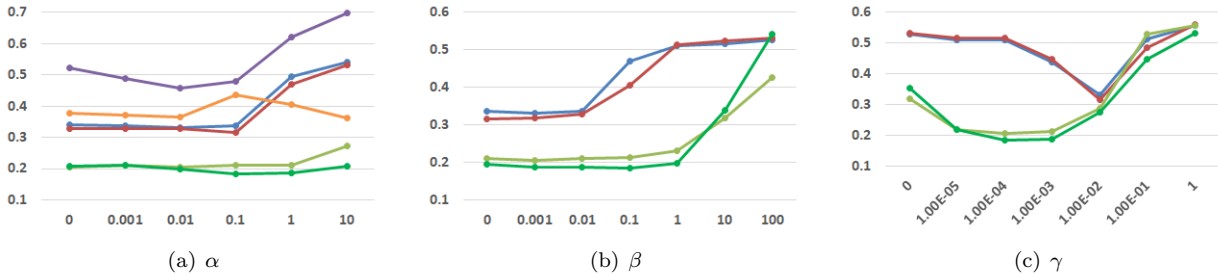

(a) $\alpha$         (b) $\beta$         (c) $\gamma$

Figure 7: Hyperparameters' ($x$-axis) sensitivity analysis based on PEHE ($y$-axis) on the **synthetic** dataset with $m_{\Gamma, \Delta, \Upsilon} = 8, m_\Xi = 1$. Legend is the same as Figure 6: purple for CFR-Net, orange for DR-CFR, blue for S-VAE-CI, red for P-VAE-CI, and light and dark green for H-VAE-CI (PB) and (CA) respectively. Plots are best viewed in color.

- Figure 7(b) shows that various $\beta$ values (*i.e.*, coefficient of KL divergence penalty) do not make much difference for H-VAE-CI (except for $\beta \geq 1$, since this large value means the learned representations will be close to Gaussian noise). We initially thought using $\beta$-VAE might help *further* disentangle the underlying factors. However, Figure 7(b) suggests that close-to-zero or even zero $\beta$s also work effectively. Our hypothesis is that the H-VAE-CI's architecture already takes care of decomposing the $\Gamma$, $\Delta$, and $\Upsilon$ factors, without needing the help of a KLD penalty. Appendix D.1 includes more evidence and a detailed discussion on why this interpretation should hold. Moreover, the KLD penalty attempts to disentangle the learned representations *within* each underlying factor; however, what we really need is the disentanglement (*i.e.*, [non-linear] independence) *across* the learned representations of different underlying factors.

- Figure 7(c) demonstrates that for a wide range, the hyperparameter $\gamma$ (*i.e.*, coefficient of the generative loss penalty), achieves the best and most stable performance in H-VAE-CI. Finding that H-VAE-CI for $\gamma \leq 0.01$ outperforms {S, P}-VAE-CI, suggests that having the generative loss term (*i.e.*, $\mathcal{L}_{\text{VAE}}$) is more important for {S, P}-VAE-CI than it is for H-VAE-CI to perform well. Note an extreme case happens at $\gamma = 0$, where the latter performs significantly (statistical) better than the former. We hypothesize that although $\mathcal{L}_{\text{VAE}}$ is helpful (note the drop in PEHE from $\gamma = 0$ to $0 < \gamma \leq 0.01$), the other terms in the objective function can *partially* impose the decomposition and learn expressive representations $Z_3$ and $Z_7$ in H-VAE-CI. This is in contrast to $Z_1$ in S-VAE-CI, and $Z_1$ and $Z_3$ in P-VAE-CI.

## 5 Future works and Conclusion

Despite the success of the proposed methods, especially the Hybrid model, in addressing causal inference for treatment effect estimation, no known algorithms can yet learn to perfectly decompose factors $\Delta$ and $\Upsilon$. This goal is important because isolating $\Delta$, and learning Context-Aware (CA) weights from it, does enhance the quality of the causal effect estimation performance — note the superior performance of H-VAE-CI (CA). The results of our ablation study in Figure 7(b), however, revealed that the currently used $\beta$-VAE does not help much with disentanglement of the underlying factors. Therefore, the proposed architectures and objective function ought to be responsible for most of the achieved decomposition. A future direction is to explore the use of better disentangling constraints (*e.g.*, works of (Chen et al., 2018) and (Lopez et al., 2018)) to see if that would yield sharper results.

The goal of this paper was to estimate causal effects (either for individuals or the entire population) from observational data. We designed three models that employ Variational Auto-Encoders (VAE) (Kingma & Welling, 2014; Rezende et al., 2014), namely Series, Parallel, and Hybrid. Each model was an improvement over the previous one, in terms of identifying the underlying factors of any observational data as well as estimating the causal effects. Our proposed methods employed Kingma et al. (2014)'s M1 and M2 models as their building blocks. Our Hybrid model performed best, and succeeded at learning decomposed representations

of the underlying factors. This, in turn, helped to accurately estimate the outcomes of all treatments. Our empirical results demonstrated the superiority of the proposed methods, compared to both state-of-the-art discriminative as well as generative approaches in the literature.

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

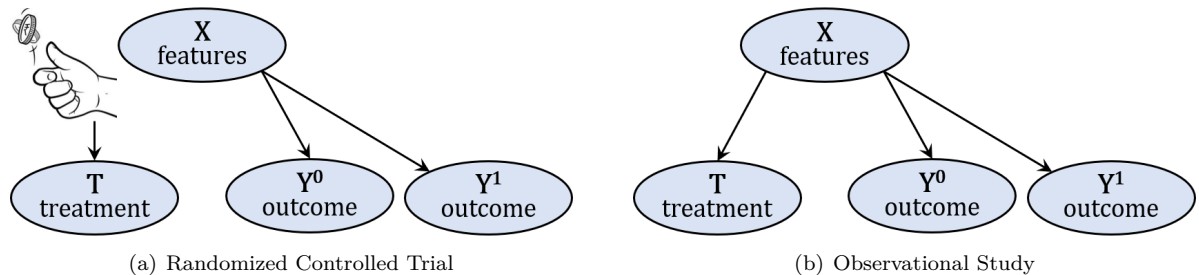

(a) Randomized Controlled Trial          (b) Observational Study

Figure 8: Belief net structure for randomized controlled trials (a) and observational studies (b). Here, $Y^0$ ($Y^1$) is the outcome of applying $T =$ treatment#0 (#1) to the individual represented by $X$.

## A   Causal Inference: Problem Setup and Challenges

A dataset $\mathcal{D} = \{ [x_i, t_i, y_i] \}_{i=1}^N$ used for treatment effect estimation has the following format: for the $i^{th}$ instance (*e.g.*, patient), we have some context information $x_i \in \mathcal{X} \subseteq \mathbb{R}^K$ (*e.g.*, age, BMI, blood work, etc.), the administered treatment $t_i$ chosen from a set of treatment options $\mathcal{T}$ (*e.g.*, {0: medication, 1: surgery}), and the associated observed outcome $y_i \in \mathcal{Y}$ (*e.g.*, survival time: $\mathcal{Y} \subseteq \mathbb{R}^+$) as a result of receiving treatment $t_i$.

Note that $\mathcal{D}$ only contains the outcome of the administered treatment (aka *observed* outcome: $y_i$), but not the outcome(s) of the alternative treatment(s) (aka *counterfactual* outcome(s); *i.e.*, $y_i^t$ for $t \in \mathcal{T} \setminus \{t_i\}$ [c0]), which are inherently **unobservable** (Holland, 1986). In other words, the causal effect $y_i^1 - y_i^0$ is never observed (*i.e.*, missing in any training data) and cannot be used to train predictive models, nor can it be used to evaluate a proposed model. This makes estimating causal effects a more difficult problem than that of generalization in the supervised learning paradigm.

Pearl (Pearl, 2009) demonstrates that, in general, causal relationships can only be learned by experimentation (on-line exploration), or running a Randomized Controlled Trial (RCT), where the treatment assignment $T$ does not depend on the individual $X$ – see Figure 8(a). In many cases, however, collecting RCT data is expensive, unethical, or even infeasible.

A solution is to approximate treatment effects from off-line datasets collected through Observational Studies. In such datasets, however, the administered treatment $T$ depends on some or all attributes of individual $X$ – see Figure 8(b). Here, as $\Pr(T \mid X) \neq \Pr(T)$, we say these datasets exhibit **selection bias** (Imbens & Rubin, 2015). Figure 9 illustrates selection bias in an example (synthetic) observational dataset. Here, to treat heart disease, a doctor typically prescribes surgery ($t=1$) to younger patients ($\bullet$) and medication ($t=0$) to older ones ($+$). Note that instances with larger (resp., smaller) $x$ values have a higher chance to be assigned to the $t=0$ (resp., 1) treatment arm; hence we have selection bias. The counterfactual outcomes (only to be used for evaluation purpose) are illustrated by small fainted $\cdot$ ($_+$) for $\neg t=1$ (0).

## B   Background

### B.1   M1 and M2 Variational Auto-Encoders

As the first proposed model, the M1 VAE is the conventional model that is used to learn representations of data (Kingma & Welling, 2014; Rezende et al., 2014). These features are learned from the covariate matrix $X$ only. Figure 10(a) illustrates the decoder and encoder of the M1 VAE. Note the graphical model on the left depicts the decoder; and the one on the right depicts the encoder, which has arrows going the other direction.

Proposed by (Kingma et al., 2014), the M2 model was an attempt to incorporate the information in target $Y$ into the representation learning procedure. This results in learning representations that separate specifications of individual targets from general properties shared between various targets. In case of digit generation, this

---

[c0]For the binary-treatment case, we denote the alternative treatment as $\neg t_i = 1 - t_i$.

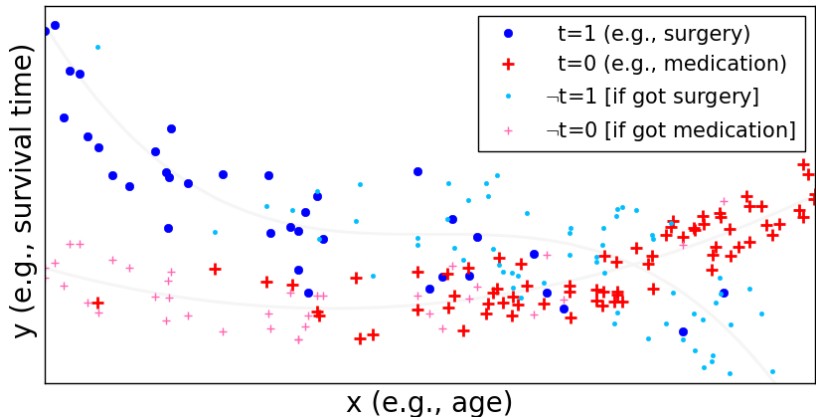

Figure 9: An example observational dataset.

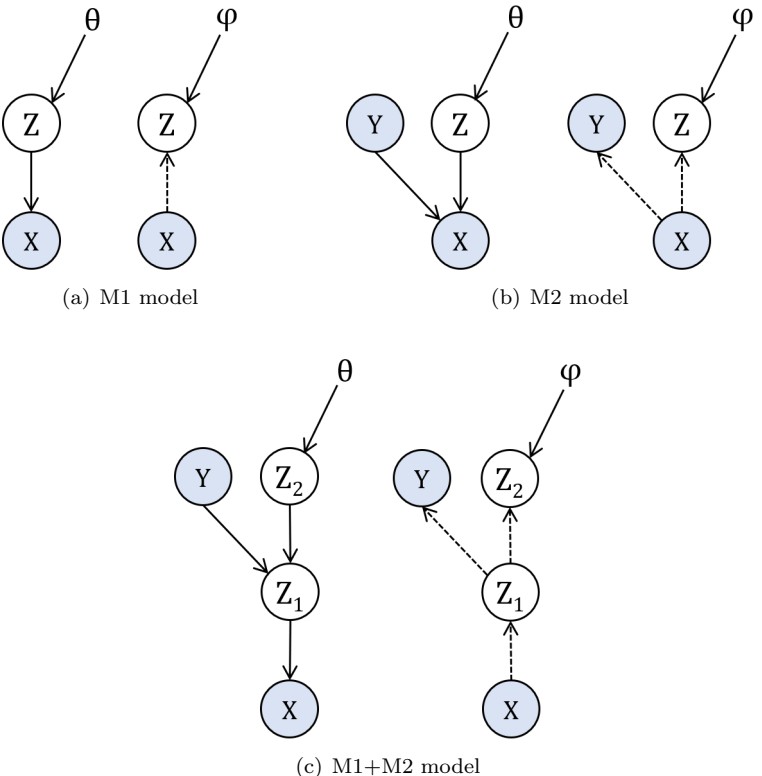

(a) M1 model

(b) M2 model

(c) M1+M2 model

Figure 10: Decoders (parametrized by $\theta$) and encoders (parametrized by $\varphi$) of the M1, M2, and M1+M2 VAEs.

translates into separating specifications that distinguish each digit from writing style or lighting condition. Figure 10(b) illustrates the decoder and encoder of the M2 VAE.

Kingma et al. (2014) found that stacking the M1 and M2 models, as shown in Figure 10(c), produced the best performance results. This way, we can first learn a representation $Z_1$ from raw covariates, then find a second representation $Z_2$, now learning from $Z_1$ instead of the raw data.

Table 2: Hyperparameters and ranges

| Hyperparameter | Range |
|---|---|
| Discrepancy coefficient $\alpha$ | $\{0, 1\text{E}\{\text{-3, -2, -1, 0, 1}\}\}$ |
| KLD coefficient $\beta$ | $\{0, 1\text{E}\{\text{-3, -2, -1, 0, 1, 2}\}\}$ |
| Generative coefficient $\gamma$ | $\{0, 1\text{E}\{\text{-5, -4, -3, -2, -1, 0}\}\}$ |

## C  Experimental Setup

### C.1  Procedure of Generating the Synthetic Datasets

Given as input the sample size $N$; dimensionalities $[m_\Gamma, m_\Delta, m_\Upsilon] \in (\mathcal{Z}^{\geq 0})^3$; for each factor $L \in \{\Gamma, \Delta, \Upsilon\}$, the means and covariance matrices $(\mu_L, \Sigma_L)$; and a scalar $\zeta$ that determines the slope of the logistic curve.

- For each latent factor $L \in \{\Gamma, \Delta, \Upsilon\}$, form $L$ by drawing $N$ instances (each of size $m_L$) from $\mathcal{N}(\mu_L, \Sigma_L)$. The covariates matrix $X$ is the result of concatenating $\Gamma$, $\Delta$, and $\Upsilon$. Refer to the concatenation of $\Gamma$ and $\Delta$ as $\Psi$ and that of $\Delta$ and $\Upsilon$ as $\Phi$ (for later use).

- For treatment $T$, sample $m_\Gamma + m_\Delta$ tuple of coefficients $\theta$ from $\mathcal{N}(0,1)^{m_\Gamma + m_\Delta}$. Define the logging policy as $\pi_0(t=1\,|\,z) = \frac{1}{1+\exp(-\zeta z)}$, where $z = \Psi \cdot \theta$. For each instance $x_i$, sample treatment $t_i$ from the Bernoulli distribution with parameter $\pi_0(t=1\,|\,z_i)$.

- For outcomes $Y^0$ and $Y^1$, sample $m_\Delta + m_\Upsilon$ tuple of coefficients $\vartheta^0$ and $\vartheta^1$ from $\mathcal{N}(0,1)^{m_\Delta + m_\Upsilon}$ Define $y^0 = (\Phi \circ \Phi \circ \Phi + 0.5) \cdot \vartheta^0 / (m_\Delta + m_\Upsilon) + \varepsilon$ and $y^1 = (\Phi \circ \Phi) \cdot \vartheta^1 / (m_\Delta + m_\Upsilon) + \varepsilon$, where $\varepsilon$ is a white noise sampled from $\mathcal{N}(0, 0.1)$ and $\circ$ is the symbol for element-wise product.

### C.2  Hyperparameters

For all CFR, DR-CFR, and VAE-CI methods, we trained the neural networks with 3 layers (each consisting of 200 hidden neurons)[c0], non-linear activation function `elu`, regularization coefficient of $\lambda$=1E-4, `Adam` optimizer (Kingma & Ba, 2015) with a learning rate of 1E-3, batch size of 300, and maximum number of iterations of $10,000$. See Table 2 for our hyperparameter search space.

## D  Further Results and Discussions

### D.1  Analysis of the Effect of $\beta = 0$

Our initial hypothesis in using $\beta$-VAE was that it might help *further* disentangle the underlying factors, in addition to the other constraint already in place (*i.e.*, the architecture as well as the discrepancy penalty). However, Figure 7(b) suggests that close-to-zero or even zero $\beta$s also work effectively. Our hypothesis is that the H-VAE-CI's architecture already takes care of decomposing the $\Gamma$, $\Delta$, and $\Upsilon$ factors, without needing the help of a KLD penalty.[c0]

In order to validate this hypothesis, we examined the decomposition tables of H-VAE-CI (similar to the performance reported in the green table in Figure 5) for extreme configurations with $\beta = 0$ and observed that they were all effective at decomposing the underlying factors $\Gamma$, $\Delta$, and $\Upsilon$. Figure 11 shows several of these tables. This means either of the following is happening: (i) $\beta$-VAE is not the best performing disentangling method and other disentangling constraints should be used instead — *e.g.*, works of Chen et al. (2018) and Lopez et al. (2018); or (ii) it is theoretically impossible to achieve disentanglement without some supervision

---

[c0]In addition to this basic configuration, we also performed our grid search with an updated number of layers and/or number of neurons in each layer. This makes sure that all methods enjoy a similar model complexity.

[c0]Therefore, it appears that we can safely drop out the KLD term altogether; which can significantly reduce the model and time complexity.

(Locatello et al., 2019), which might not be possible to provide in this task. Exploring these options is out of the scope of this paper and is left to future work.

| H-VAE-CI | | | | | | | |
|---|---|---|---|---|---|---|---|
| | Z1 | Z2 | Z3 | Z4 | Z5 | Z6 | Z7 |
| Γ | 0.2181 | 0.2185 | **1.8791** | **1.7711** | 0.2164 | 0.2190 | 0.2039 |
| Δ | 0.6041 | 0.6051 | 0.6142 | 0.6308 | **0.8688** | 0.8315 | **0.6138** |
| Υ | **0.8523** | 0.8552 | 0.3321 | 0.3834 | 0.7552 | 0.7859 | **0.7384** |

| H-VAE-CI | | | | | | | |
|---|---|---|---|---|---|---|---|
| | Z1 | Z2 | Z3 | Z4 | Z5 | Z6 | Z7 |
| Γ | 0.2242 | 0.2182 | **0.7439** | 0.6770 | 0.2373 | 0.2583 | 0.2189 |
| Δ | 0.3014 | 0.2963 | 0.2612 | 0.3169 | **0.6051** | 0.5845 | **0.7048** |
| Υ | **0.5385** | 0.5430 | 0.3211 | 0.3303 | 0.4394 | 0.4412 | **0.4571** |

| H-VAE-CI | | | | | | | |
|---|---|---|---|---|---|---|---|
| | Z1 | Z2 | Z3 | Z4 | Z5 | Z6 | Z7 |
| Γ | 0.4254 | 0.4493 | **0.7090** | 0.6872 | 0.3823 | 0.3771 | 0.3874 |
| Δ | 0.6438 | 0.6461 | 0.2750 | 0.3129 | **0.7452** | 0.7569 | **0.8237** |
| Υ | **0.7760** | 1.1200 | 0.3137 | 0.3480 | 0.7240 | 0.7464 | **0.6717** |

| H-VAE-CI | | | | | | | |
|---|---|---|---|---|---|---|---|
| | Z1 | Z2 | Z3 | Z4 | Z5 | Z6 | Z7 |
| Γ | 0.3646 | 0.3643 | **1.1457** | 0.8659 | 0.3942 | 0.4069 | 0.3166 |
| Δ | 0.5127 | 0.5307 | 0.6463 | 0.5794 | **0.7016** | 0.6717 | **0.7652** |
| Υ | **0.5565** | 0.5780 | 0.4260 | 0.3964 | 0.4119 | 0.4234 | **0.4534** |

| H-VAE-CI | | | | | | | |
|---|---|---|---|---|---|---|---|
| | Z1 | Z2 | Z3 | Z4 | Z5 | Z6 | Z7 |
| Γ | 0.8821 | 0.8752 | **0.5805** | 0.5782 | 0.3326 | 0.3309 | 0.4006 |
| Δ | 1.2542 | 1.2480 | 0.2843 | 0.3488 | **0.8553** | 0.8568 | **0.9392** |
| Υ | **1.914** | 1.923 | 0.4498 | 0.4797 | 0.7791 | 0.7757 | **0.7969** |

| H-VAE-CI | | | | | | | |
|---|---|---|---|---|---|---|---|
| | Z1 | Z2 | Z3 | Z4 | Z5 | Z6 | Z7 |
| Γ | 0.0850 | 0.0875 | **1.8791** | **1.7711** | 0.1464 | 0.1459 | 0.1833 |
| Δ | 0.6107 | 0.6085 | 0.6142 | 0.6308 | **0.7851** | 0.7937 | **0.6878** |
| Υ | **0.8349** | 0.8242 | 0.3321 | 0.3834 | 0.5177 | 0.5073 | **0.6832** |

Figure 11: Decomposition tables for H-VAE-CI with $\beta = 0$.

