# OpenReview forum: "Variational Auto-encoders for Causal Inference"
_TMLR — Rejected by TMLR_

### Review · Reviewer_sCX6 · 2022-08-24

**Summary Of Contributions:**

This paper introduces three models built upon variational auto-encoders that decompose observational data $\{X_i, T_i, Y_i \}^N_{i=0}$ into disentangled latent variables from which we can estimate treatment effects. The paper considers a decomposition of the covariate variables $X$ into four independent factors $\Gamma, \Delta, \Upsilon, \Xi$, where the treatment T is independent of $\Upsilon, \Xi$ and the interventional outcome $Y$ is independent of $\Gamma, \Xi$. $\Xi$ represents the information in $X$ that does not impact the outcome $Y$ nor the treatment selection, $\Upsilon$ the information in $X$ that only impacts the outcome, $\Delta$ the information in $X$ that impact both the treatment attribution and the outcome, and $Gamma$ the information in $X$ that only impacts the treatment attribution. The authors suggest that the proposed models can learn a latent representation $Z_1$ that contains only the information $\{\Delta, \Upsilon\}$ in $X$ that is relevant to evaluating the interventional probability of the treatment $P_T(Y^T | X)  := P_T(Y^T | \{\Delta, \Upsilon\})$. Experiments on two real-world and one synthetic dataset suggest that the proposed algorithm can disentangle the latent representations appropriately and lead to better treatment effect estimation than concurrent methods.

**Broader Impact Concerns:**

I do not see any direct ethical concern regarding the methods presented.

**Requested Changes:**

- I strongly encourage the authors to improve the writing of the manuscript. In particular, Section is tough to follow at the moment. The causal inference setting in Appendix A should probably go back to the main text before describing the methods. I would also suggest introducing all the symbols before describing existing algorithms. For example, $\Delta$ is mentioned once before its formal definition. It is also the first time I have seen the term M1 and M2 VAEs, it does not seem to be a widespread denomination for such models, and I would suggest finding better names.
- It would be nice to provide a more formal motivation of why the VAEs could eventually learn the right disentanglement.
- Explaining formally why the different models will potentially learn the right disentanglement would help me get convinced of the approach.

**Strengths And Weaknesses:**

# Strengths
- Building disentangled representation with specific independence assumptions with a variational auto-encoder is an excellent idea.
- The empirical validation of the method is convincing and shows a real benefit of the newly introduced models compared to alternative strategies.

# Weaknesses
- My main concern is about the quality of the writing. I found the manuscript very hard to read, and I must acknowledge that I probably misunderstood parts. The related work is particularly hard to understand, and I would encourage the authors to improve their descriptions of the existing methods. I also found most equations very poorly formatted (e.g. strong ignorability and ELBO on two lines without a proper justification...)
- I did not understand why $Z_1$ should be independent of $T$ as it contains information about $\Delta$ that is itself not independent of $T$. However, the disc loss should indeed help there. But why do you then give $T$ to the posterior distribution of $Z_1$ if they are supposed to be independent.
- The motivation for using a generative rather than a discriminative approach did not convince me. I can't entirely agree with the statement about the overfitting and confidence issues of discriminative methods that generative approaches would magically solve. The statement about Bayesian inference seems orthogonal to what is done in the paper. I encourage the authors to reformulate the benefit of a generative approach over a discriminative one.
- It would be worth giving more details about the datasets.
- Figure 7c is oddly rendered.

---

> ### Author Response · Authors · 2022-09-17
> **Response to Reviewer sCX6**
>
> We thank the reviewer for their detailed feedback. We have updated the manuscript (in colored text, marked with the reviewer’s ID) to incorporate the requested changes.
>
> To explain why we want to make $Z_1$ independent of $T$: this is because we want to remove the $\Gamma$ part of the input $X$ from $Z_1$’s learned representation. For the series and parallel models, however, $Z_1$ should also contain $\Upsilon$ as this is the only node that contributes to the posterior of $Y$ (i.e., recall $q(Y|Z_1)$). For the hybrid model, however, $Z_1$ would only include $\Upsilon$ as a result of including the discrepancy term in the objective function since we have another node $Z_5$ to capture $\Delta$.
>
> To explain why $T$ is given in the posterior of $Z_1$, note that this is only true for the series and parallel models. As explained in the previous paragraph, $Z_1$ in the series and parallel models would (and should) also capture $\Upsilon$ as well as $\Delta$; therefore, information of $T$ is necessary. In the Hybrid model, however, $T$ is no longer necessary for the posterior of $Z_1$ (i.e., $q(Z_1|Z_7)$ does not involve $T$) as it is only supposed to capture $\Upsilon$.
>
> To motivate why we want to use a generative model as opposed to a discriminative one: learning the underlying data generating process is quite important for causal inference due to the interventional queries that we need to make here. That is, we need to know how the data was generated in order to accurately estimate what would have happened had a certain variable had taken a different value. We have added this point to the respective paragraph in the manuscript (in orange text, on page 2).
>
> We have added more details to the descriptions of the IHDP and ACIC’18 benchmarks in the manuscript (in green text, on page 9). The details of our synthetic benchmark is elaborated in Appendix C.1.
>
> We have modified Sec. 4.2 (in orange text, on pages 9 and 10) to make it clearer.
>
> We have added parts of the Appendix A on causal inference setting to the main text (in purple text, on page 1) in order to make the paper more self-contained.
>
> We have added a brief description of all underlying factors (including $\Delta$) to the caption of Fig. 1 (in green text, on page 3).
>
> The names for the M1 and M2 models are coined by their inventors Kingma, et. al in their NeurIPS 2014 paper. We are not changing them for consistency.
>
> Regarding “a more formal motivation of why the VAEs could eventually learn the right disentanglement” and “why the different models will potentially learn the right disentanglement”, we first refer to the original paper by (Kingma et. al, 2014), which proposed the building blocks M2 and M1+M2 VAEs that we use in our models. These VAEs facilitate the target variables to provide guidance in learning representations that are more effective in the downstream prediction task. Note that only the hybrid model can learn the right complete disentanglement of the underlying factors. We hypothesize that this is due to the extra supervision from $Y$ and $T$, facilitated by the use of M2 and M1+M2 VAEs, as shown empirically in Sec. 4.2. We have updated the manuscript (in green text, on page 5) to indicate this point.

---

### Review · Reviewer_BLq7 · 2022-08-25

**Summary Of Contributions:**

This paper proposes three latent-based models for estimating causal effects from observational data. The three methods are compared against a variety of competing methods on two existing datasets and one synthetic dataset.

**Requested Changes:**

The authors must clarify their motivation, lay down the assumptions required for their method to work, clarify how their method goes from the estimation of a latent-based model of $p(x|y,t)$ to an estimate of $p(y|x,do(t))$, and express the limitations of their method.

### Detailed reading comments:

p2: Generative models based on Bayesian inference, on the other hand, can handle both of these drawbacks -> Bayesian inference can also be applied to discriminative models. It seems your point here is that generative models are superior to discriminative models, due to Bayesian inference. The reasoning here simply does not hold.

p2: Contributions -> the contribution is unclear to me. Does Bayesian inference address the problems of unobservability or selection bias introduced earlier ? How so ?

p2:  our proposed Hybrid model is the best [...] of any observational dataset -> How is this shown ? Do you have proof of that statement ? You can't perform experiments on all possible observational datasets.

p3: it is known that only generative models [...] can truly identify the underlying data generating mechanism -> I fail to see how that is related, or why that would be an issue. In the end causal inference is about estimating a discriminative model, $p(y|x,do(t))$. I do not see how infering a generative model as an intermediate step towards that goal would be a requirement.

p3: this is a step in the right direction -> how so? Why would using VAEs be the right direction?

p4: we assume that $X$ follows an unknown joint probability distribution [...] independent factors. -> What is the causal structure between X and the independent factors? Is there a reason why the arrows in Figure 1 remain undirected? Are the factors are a cause of $X$? This would indeed make the factors independent. Otherwise, why would the factors be independent?

p3: the belief net in Figure 1 [...] degenerate. -> I am not convinced by that statement. Here you assume that $X$ cannot be a common effect of $T$ and $Y$. This is a strong assumption. Using $X$ as an adjustment variable in that case would be dangerous, as it would bias the estimated causal model.

p4: even in the ones that have degenerate factors -> What is meant here by degenerate? A proper definition is missing.

p5 Posteriors: assuming you want to model $p(x|t,y)$ with the latent model that follows the architecture (a), the encoder should be $q(z_1,z_2|x,y,t) = q(z_1|x,t,y)q(z_2|y,z_1)$. Here in the posteriors you list $q(z_1|x,t)$, $q(y|z_1)$ and $q(z_2|y,z_1)$. What is the term $q(y|z_1)$ doing here? It doesn't even appear in equations (1) or (2). Why do you have $q(z_1|x,t)$ instead of $q(z_1|x,t,y)$? I can't follow the reasoning here.

p5 eq1: $\log p(x_i|t_i,y_i)$ -> so in the end you train a discriminative model? What was your point about generative vs discriminative methods in earlier sections then?

p5 eq2: This equation is ill-defined. Should I assume $x$ to be $x_i$? Why does the subscript disappear sometimes? I suppose $KL(q(z_2|y,z_1) || p(z_2))$ is the expected KL-divergence. Which distribution of $y$ and $z_1$ is used to compute the expectation? Same comment for all KL terms in equations 2, 4 and 6. Furthermore, using subscripts to denote different random variables ($z_1,z_2$) and / or their realization ($x_i,y_i$) is confusing.

p5: the other distributions are parameterized as deep neural networks -> how? Do you also use multivariate Gaussian distributions? Details are missing.

p6 3.1.3: It is really unclear to me how the hybrid model relates to the hypothetical ground-truth generative model from Figure 1. Why not simply building a latent model that follows exactly the architecture from Figure 1?

p7 3.3 Discrepancy: I fail to see how $T$ and $Z\_1$ can be dependent in the learned model. In the end, your model $p(x,z|y,t)$ follows one of the architectures described in Figure 3. Then, the constraint $T \\perp Z\_1$ is enforced by construction.

p7 3.4 Predictive Loss: It is unclear here how the weights $w_i$ are computed. It is unclear also over which distribution the expectation for $\hat{y}_i$ is computed. Does $z_1$ come from the prior or the encoder model? Do you refer here to $Z_1$ from model (a), (b) or (c)?

p9 4.2.1: This whole section is extremely unclear. What is a trained network $Z_j$? Why are zeroes filled in the dataset? What is an elu output? What is $O_{i,j}$, with respect to $X$, $Y$, $T$ and $Z$ discussed in earlier sections? How does any of this relate to the problem of causal inference?

p9 4.3: Individual-based: how can this measure be computed on the IHDP dataset? It seems that this measure requires knowledge of the effect of treatment (t=1) and no treatment (t=0) on each individual. This information is not available in randomized control trials.

**Strengths And Weaknesses:**

### Strengths

I found the litterature review interesting. The paper exposes an overview of the field, and discusses 6 recent methods for causal inference from observational data: CFR-Net (Shalit et al., ICML 2017), DR-CFR (Hassanpour & Greiner, ICLR 2020), Dragon-Net (Shi et al., NeurIPS 2019), GANITE (Yoon et al., ICLR 2018), CEVAE (Louizos et al., NeurIPS 2017), TEDVAE (Zhang et al., AAAI 2021).

The reported performance improvements (Table 1) seem significant.

### Weaknesses

My biggest concern is clarity and soundness.

 - The exposition of the proposed method is very poor. Why are you learning models of the distribution $p(x|y,t)$? How do you extract the causal distribution $p(y|x,do(t))$ from your models?

 - The assumptions and limitations of the proposed method, and also of the competing ones, is missing. The task of estimating causal effects from observational data only is in general impossible. What are the assumptions that make the proposed method correct? What is the intuition behind the proposed latent-based method? Why would the disentanglement latent variables help improve the quality of the final causal estimator?

 - The connection between generative models, discriminative models, Bayesian methods and causal inference is poorly motivated, and remains very unclear to me.

 - how the evaluation metrics are computed is unclear. How can responses to treatment and non-treatment ($y_i^1$ and $y_i^0$) be available for each individual of a randomized trial?

 - Section 4.2 is extremely ambiguous, I don't understand the setting or the purpose of the experiment.

At this point I understand neither the motivation behind the proposed method, how it works, or the limitations of the method. I also have concerns about the reported experimental performance, as I don't fully understand how the metrics can be computed.

---

> ### Author Response · Authors · 2022-09-17
> **Response to Reviewer BLq7 [1/2]**
>
> We thank the reviewer for their detailed feedback. We have updated the manuscript (in colored text, marked with the reviewer’s ID) to incorporate the requested changes.
>
> We try to maximize the lower bound on the likelihood of the observed data (i.e., $P(X|Y, T)$), in order to find the underlying data generating process from which our data is sampled. This is the generative part of the algorithm. After learning the representations of the underlying factors, we then use a discriminative approach to estimate the outcomes. We have added a footnote (in blue text, one page 6) to point this out.
>
> We have updated the manuscript to explicitly mention the assumptions that we make in this work (in teal text, on page 2). The other methods described in the literature review also make the same three assumptions.
>
> There are several reasons why learning disentangled representations of the non-overlapping underlying factors would improve the quality of the final causal estimator: (i) by knowing $\Gamma$, we can discard it and so remove the reducible part of the selection bias; (ii) by knowing $\Delta$ (that is not polluted with $\Upsilon$), we can learn better context-aware weights (based on importance sampling) that can account for the remaining non-reducible selection bias.
>
> We have updated the paragraph on the use of generative vs. discriminative model (in orange text, on page 2); and elaborated on our motivation of employing a generative model as opposed to a discriminative one for the purpose of representation learning. Briefly, our intuition is that learning the underlying data generating process is quite important for causal inference due to the interventional queries that we need to make here. We thank the reviewer for raising this question as answering it helped make the contribution of this work clearer.
>
> Sec. 4.2 shows that the proposed models (especially the hybrid model) have the capability to learn disentangled representations of the underlying factors. Sec. 4.3 then shows that they are effective in estimating the causal effects, not only in public benchmarks, but also in a synthetic benchmark of datasets that exhibit various degrees of selection bias (i.e., various dimensionality of the underlying factors).
>
> Regarding the arrows in Fig. 1, $X$ is in fact composed of $\Gamma$, $\Delta$, and $\Upsilon$ –i.e., their concatenation makes $X$. Therefore, every feature belongs to exactly one of these. We have indicated this point in the caption of Fig. 1 (in blue text, on page 3).
>
> We do not assume $X$ cannot be a common cause of $Y$ and $T$. In fact, the $\Delta$ factor in $X$ includes the confounding variables between $Y$ and $T$.
>
> We say a factor is “degenerate” if its dimensionality is zero, as this effectively means it does not appear in $X$. We have updated the manuscript (in blue text, on page 5) to include this definition.
>
> You are correct: the posterior of the series model is indeed $q(z_1, z_2, y | x, t)$; however, we assume it can be factorized as $q(z_1 | x, t) \times q(z_2 | z_1, y) \times q(y | z_1)$ (this is a common assumption in the literature; for example this is implied by the supervised Evidence Lower BOund (ELBO) in Eq. (3) of (Louizos et al., 2015)). Since $Y$ is a descendant of $Z_1$ in the encoder (i.e., variational posterior parametrized by $\phi$), it is not given in $q(z_1 | x, t)$.
>
> The posterior $q(y | z_1)$ does not appear in the ELBO because we only work with the observed outcomes $Y$ and do not use the unsupervised part of the M2 objective. This is because the unsupervised part is supposed to extract useful information from those $X$ that are not accompanied by a target value $Y$ (i.e., unsupervised) by imputing their respective $Y$s. However, our unsupervised $X$ are exactly the same as the supervised $X$, except that in the former, the counterfactual outcomes are missing --- hence, we don’t expect including the unsupervised loss to have any added value). As a result, we do not require a prior on $Y$ and its respective KL term between the posterior and prior in the ELBO.
>
> We do train a discriminative model in the end, but only after we have learned the underlying factors by a generative model. Our empirical results show that a generative approach for representation learning yields better performance in the downstream discriminative task of causal inference compared to a discriminative approach for representation learning as proposed in (Hassanpour and Greiner, 2020).
>
> The KL term in Eqs. (2), (4), and (6) is calculating the KL divergences between the entire probability distributions of a posterior and a prior, not single data points. That is why we do not need a subscript data-point counter for the KL terms.
>
> Following the literature, all distributions are assumed to be multivariate Gaussians with diagonal covariance (Kingma & Welling, 2014). We have updated the manuscript (in blue text, on page 6) to indicate this.

---

> > ### Author Response · Authors · 2022-09-17
> > **Response to Reviewer BLq7 [2/2]**
> >
> > Regarding “why not simply building a latent model that follows exactly the architecture from Fig. 1”: we use the M2 and M1+M2 building blocks in our proposed models due to their capability of learning representations that are guided by the targets $T$ and $Y$; thus allowing us to obtain disentangled representations of the underlying factors. We are not sure how to provide such guidance with the belief net in Fig. 1.
> >
> > Regarding the dependence of $T$ and $Z_1$: while the belief nets enforce dependence, since one of $Z_1$’s parents is $Y$, and $Y$ is either the outcome of treatment 1 or 0, we have an information leak from $T$ to $Z_1$ through $Y$, which we should remedy in order to avoid learning the $\Gamma$ factors in $Z_1$. We have added a footnote (in blue text, on page 8) to elaborate on this.
> >
> > The paper considers 2 approaches from the literature for derivation of the weights. Shalit et. al. (2017)’s approach is described in the CFR-Net paragraph in Sec. 2. These weights are based on the population-based probability distribution of $T$. Hassanpour and Greiner (2019)’s approach, on the other hand, is context-aware in that it uses the learned representations to derive the weights following the importance sampling principle.
> >
> > We use $Z_1$ (which captures both $\Delta$ and $\Upsilon$) for deriving the weights for the series and parallel models and $Z_5$ (which captures $\Delta$) for the hybrid model. We have added a footnote (in blue text, on page 8) to elaborate on this point.
> >
> > There was a typo in $\hat{y_i}$; we meant to set $\hat{y_i}$ to be the mean of the $q_{\phi}^{t_i}(y_i | z_{1_i})$ distribution. We thank the reviewer for pointing this out. We have fixed this in the manuscript (in blue text, on page 8).
> >
> > To explain the Sec. 4.2, the trained network $Z_j$ is basically a neural network that yields the $\mu$ of the respective node in the belief nets in Fig. 3 given its inputs. We fill the location of each underlying factor with “1”s, and elsewhere with “0”s in order to measure how much of this dummy signal could pass through different representation networks. For example, if a representation network $Z_j$ is supposed to capture $\Delta$, we expect to see the output of $Z_j$ network to be high for input of $V_2$ and low for inputs of $V_1$ and $V_3$ dummy variables. We let $O_ij$ refer to the output of network $Z_j$ (with elu as its activation function of the last layer) for input $V_i$. We use this process to evaluate the quality of learning disentangled representations of the underlying factors; note it does not directly evaluate the quality of the proposed methods in terms of causal effect estimation. We have modified Sec. 4.2 (in orange text, on pages 9 and 10) to make this process clearer.
> >
> > Regarding how individual-based evaluation criterion can be calculated on our datasets, please note that ALL datasets used in evaluating causal inference approaches include the counterfactual outcomes. Of course, the learning procedures do not have access to that information; we only use the counterfactuals for the purpose of evaluation. As Mentioned in the beginning of Sec. 4.1, “Researchers often evaluate treatment effect estimation on semi- or fully- synthetic datasets that include both factual and counterfactual outcomes.” The IHDP and ACIC’18 datasets are semi-synthetic, where the covariates are taken from a real-world experiment, but the outcomes of both treatments are synthesized to allow for proper evaluation of the proposed causal effect estimation methods.

---

> > > ### Comment · Reviewer_BLq7 · 2022-09-22
> > > **Not convinced**
> > >
> > > Thank you for these answers.
> > >
> > > Despite the explanations and the changes made to the paper, clarity and soundness remain big concerns of mine.
> > >
> > > Moreover, the reasoning of the paper makes no sense to me. The assumptions, no unobserved confounder and strict positivity, make the problem of causal inference from observational data trivial. To estimate the causal effect $p(y|x,do(t))$ it suffices to estimate the distribution $p(y|x,t)$. Then the only remaining problem is that of statistical learning, which can be done using any off-the-shelf supervised learning algorithm. I feel that the claims made in the paper, addressing the problem of causal inference from observational data, is not accurate nor convincing. Under these assumptions, all supervised learning method address the problem of causal inference.

---

### Review · Reviewer_W8Yz · 2022-09-06

**Summary Of Contributions:**

The paper develops "bayesian" methods for estimating causal effects. The idea is to disentangle the different factors that generate X, Y, T etc, which then help effect estimation. The three methods built in the paper have different variables that X,Y,T disentangle into and this seems to help estimate effects well.

**Broader Impact Concerns:**

The ethical implications involve the same as any causal estimation method. It can be used for improving or worsening many existing inequities in society. However, the method does not build large-scale models that could easily be exploited, so I see no immediate issues.

**Requested Changes:**


The following I think are important to address to make the paper a valuable addition to effect estimation:

1. Please make the assumptions in the paper very clear; where is ignorability assumed? In general, it seems like it does not hold with X.
2. If ignorability does not hold, what is X? Is it a proxy (downstream variable to unobserved confounder)? There is a giant literature of proxy methods that is not discussed in the paper. This be addressed by including discussion and potentially even comparisions. See https://arxiv.org/abs/2103.14029 and related work.
2. The belief nets cannot be causal graphs (if they were, intervening on T does not change the graph which makes the problem trivial). What is the point of the belief nets with the particular directions of arrows for example?
4. It is unclear to me why the disentangling works that differently in the three models, especially because disentanglement is not identifiably in general: https://arxiv.org/abs/1907.04809. It seems to me that this is due to the additional supervision provided in the way that the representations connect to each variables. This should be surfaced earlier.

**Strengths And Weaknesses:**



Strengths: the experiments are impressive. The presentation is good but could be improved to make sure that causality part is better discussed, including assumptions etc. The related work is well-done.


Weaknesses: the paper lacks discussion on what makes their method work better than existing work; expressiveness of representations is insufficient to improve causal effect estimation. This sentence "This additional information helps the model to learn more expressive representations, which was not possible with the unsupervised M1 model." seems to hold the right intuition. Expanding on this during the motivation will help better understand the key value of the presented methods. The assumptions necessary to guarantee identifiability are missing.

---

> ### Author Response · Authors · 2022-09-17
> **Response to Reviewer W8Yz**
>
> We thank the reviewer for their detailed feedback. We have updated the manuscript (in colored text, marked with the reviewer’s ID) to incorporate the requested changes.
>
> Yes, this work does assume ignorability, along with SUTVA and overlap. We have updated the manuscript to explicitly mention these assumptions (in teal text, on page 2). Also, ignorability holds since if $X$ is given, in effect $\Delta$ is given, and therefore, $Y \perp T | X$.
>
> Yes, we agree that the belief nets in Fig. 3 are not causal graphs. Instead, they are belief nets for the respective VAEs that we have designed to learn the underlying factors and estimate the treatment effects.
>
> Regarding why disentangling works in the proposed architectures, we hypothesize that the extra supervision from $Y$ and $T$, facilitated by the use of M2 and M1+M2 VAEs, is responsible for this disentanglement. In other words, in the proposed architectures, $Y$ and $T$ guide each of the representation nodes to capture the appropriate factor. We thank the reviewer for this great insight; we have updated the manuscript to indicate this point (in green text, on page 5).

---

### Decision · Action_Editors · 2022-10-15

**Recommendation:** Reject

**Comment:**

In addition to concerns about evidence, the reviewers also had concerns about clarity and presentation of ideas.  Specifically the discussion of generative models, discriminative models, Bayes and causal inference as motivation confused the reviewers.

Quoting a specific reviewer:

> My suggestion is that either the authors design and implement ablations, or give intuition (maybe in a simple synthetic example.) about why they believe their method works, especially because they already assume that positivity (overlap) and ignorability are satisfied.

Additional evidence and clarity will make this submission much more compelling.

**Audience:**

The reviewers agree that TMLR is an appropriate venue for this work.

**Claims And Evidence:**

While the reviewers viewed the experimental results favorably, they also agreed that they shed little light on what exactly about the new method improves upon baselines.  In this sense, the evidence isn't clear.  Reviewers also pointed out some statements made that were either vague or unsupported by experimental evidence.  Specifically, one reviewer points to

> "While the idea behind DR-CFR provides an interesting intuition, it is known that only generative models (and not discriminative ones) can truly identify the underlying data generating mechanism. This paper is a step in this direction."

and

> ", the authors found that TEDVAE is not successful in identifying the risk factors zy (equivalent to our Υ). This might be because their model does not have a mechanism for distinguishing between the risk factors and confoundings zc (equivalent to our ∆). The evidence is in their Equation (8), which would allow zy to be degenerate and have all information embedded in zc. Our work, however, proposes an architecture that can achieve this decomposition"

as statements that can be supported by carefully designed synthetic experiments with ablations.